# Combined Programmed Intermittent Bolus and Patient-Controlled Bolus Is a More Favorable Setting for Epidural Pain Relief Than Continuous Infusion

**DOI:** 10.3390/healthcare11091350

**Published:** 2023-05-08

**Authors:** Shih-Kai Liu, Shao-Chun Wu, Shao-Chi Hung, Kuen-Bao Chen, Amina M. Illias, Yung-Fong Tsai

**Affiliations:** 1Department of Anesthesiology, China Medical University Hospital and China Medical University, Taichung 404327, Taiwan; u402053@gmail.com (S.-K.L.);; 2Department of Anesthesiology, Kaohsiung Chang Gung Memorial and College of Medicine, Chang Gung University, Kaohsiung 833401, Taiwan; shaochunwu@gmail.com; 3Department of Anesthesiology, Linko Chang Gung Memorial Hospital and Graduate Institute of Clinical Medical Sciences, College of Medicine, Chang Gung University, Taoyuan 333423, Taiwan

**Keywords:** continuous epidural infusion, epidural analgesia, maternal satisfaction, patient-controlled epidural analgesia, programmed intermittent bolus

## Abstract

Epidural analgesia is a suitable and effective treatment for labor pain. However, the preferable modality setting for delivery remains debatable. This study adopted a programmed intermittent epidural bolus (PIEB) setting in conjunction with a patient-controlled epidural analgesia (PCEA) setting to improve the quality of labor analgesia and reduce the number of medical staff. We conducted a prospective observational analysis of primigravida parturients scheduled for spontaneous labor, which required epidural analgesia for painless labor. A total of 483 healthy primigravida parturients with singleton pregnancies were included in this cohort; 135 nulliparous patients were assigned to the continuous infusion setting (CEI) group and 348 to the PIEB + PCEA group. Compared to the CEI setting, the PIEB + PCEA setting significantly reduced the manual rescue by the clinician, extended the time required for the first manual rescue dose, and acclaimed good maternal satisfaction. The use of the CEI mode increased for poor performance requiring more than two rescues with an odds ratio of 2.635 by a binary logistic regression analysis. Using the PIEB + PCEA setting as the maintenance infusion had a longer duration for the first requested manual rescue, fewer manual rescue boluses, excellent satisfaction, and no significant increase in adverse events compared to the CEI setting.

## 1. Introduction

Epidural analgesia is considered one of the most effective methods of labor analgesia [1]. The modality settings of intermittent boluses and continuous epidural infusion (CEI) are commonly used to maintain labor analgesia. In addition, patient-controlled epidural analgesia (PCEA) provides an optional and timely method for parturients to trigger analgesia whenever necessary [2]. A combination of different settings was used to develop high-quality analgesia and reduce undesirable side effects. Programmed intermittent epidural bolus (PIEB) is an emerging approach to treat labor pain. It is characterized by automated boluses at regular intervals and injection volumes and is believed to provide high infusion pressure for the efficient spread of anesthetics in the epidural space [3,4]. The determination of the optimal arrangement of intermittent volume and time intervals for PIEB settings has been studied for many years, and some studies have shown good performance and satisfaction [5,6,7]. Fidkowski et al. reported that PIEB regimens at 10 mL every 60 min for labor analgesia decreased breakthrough pain and manual rescue compared to CEI [8]. Other studies revealed that PIEB resulted in lesser anesthetic consumption [6,7,9,10,11,12], fewer manual rescues by clinicians [5,6,9,11,13], or sustained duration of analgesia [9,14] compared to CEI, but not all benefits could be confirmed simultaneously in a single study. Therefore, the manipulation of the regimen and setting is still a topic of interest.

Enduring labor pain could be a nightmare and tremendous physiological trauma for some parturients. The PIEB, CEI, and PCEA settings were designed to reduce these negative impacts on parturients. Breakthrough pain in labor analgesia is a concern for both doctors and patients. Prevention of the occurrence of breakthrough pain and, hence, fewer clinician rescues to improve the quality of analgesia are critical to maternal satisfaction. Theoretically, an adequate volume of anesthetic should be instilled into the epidural space to achieve sufficient spread with good coverage of sensory transmission before breakthrough pain occurs. Less intervention with manual boluses and PCEA triggers probably indicates a high quality of labor analgesia. Wong et al. designed various manipulations of PIEB settings and found that good performance was correlated with bolus volume, intermittent interval, and anesthetic consumption [15]. However, the quality of analgesia, including visual analog scale (VAS), time to the first PCEA or manual bolus by clinicians, and number of manual boluses, was not significantly different between the settings in their results.

This study adopted a PIEB setting using a commercial pump (CADD Solis Epidural Pump, Smiths Medical, St. Paul, MN, USA), combined with a PCEA setting, to improve the quality of labor analgesia and reduce the number of medical staff. This prospective study aimed to evaluate the benefits of the PIEB + PCEA dual setting in relieving labor pain. We hypothesized that the PIEB + PCEA setting would reduce the number of manual boluses, which is a result of prolonging the time to the first manual rescue dose (T_1st_) and improving the quality of analgesia (VAS and maternal satisfaction) compared to the CEI setting.

## 2. Methods

### 2.1. Study Participants

This prospective observational study was conducted on full-term primigravida parturients scheduled for vaginal delivery, which required epidural analgesia through a pump device for labor pain control between April 2019 and March 2020. This study was approved by the Research Ethics Committee of China Medical University and Hospital (registration number: CMUH108-REC1-022) and Research Registry (researchregistry8047; https://www.researchregistry.com/browse-the-registry#home/registrationdetails/62b9bed8bbbe0c001e327577/) accessed on 27 June 2022. and was conducted following the Declaration of Helsinki. Our study also complied with the Strengthening the Reporting of Observational Studies in Epidemiology statement guidelines [16]. Written informed consent was obtained from each primigravida parturient before epidural pain control. Parturients were healthy without systemic disease before becoming pregnant, and only epidural analgesia delivered through a pump device was included. Participants with a history of long-term opioids or painkillers or who experienced preterm delivery (gestational age <37 weeks), unexpected cesarean section, or combined spinal-epidural analgesia (CSE) were excluded from this study. In total, 483 healthy primigravida parturients with singleton pregnancies were included in this cohort.

### 2.2. Modality Settings of the Delivery Device for Labor Analgesia

All recruited participants received epidural analgesia for labor pain relief after intravenous infusion with 500 mL of normal saline for volume expansion. After the placement of an epidural catheter, which was 19-gauge sized with multiple orifices, a 10 mL bolus of the loading dose containing 2 mg/mL of bupivacaine and 5 μg/mL of fentanyl was immediately prescribed to the participants. For the maintenance dose, 0.625 mg/mL of ropivacaine and 2 μg/mL of fentanyl were administered by the delivery pump after 1 h. Participants’ allocation was determined based on the attending anesthesiologist responsible for obstetric anesthesia on the day of delivery, as well as the progress of labor, individual experience, and medical preferences. Two delivery device modalities were adopted for the labor analgesia: CEI mode (control group) and PIEB + PCEA mode (study group) via a commercial pump (CADD Solis Epidural Pump, Smiths Medical, St. Paul, MN, USA). In the control group, the maintenance delivery rate was fixed at 8 mL/h in participants with body height < 160 cm. In contrast, the delivery rate was set at 10 mL/h for people with body height ≥ 160 cm. In the study group, a maintenance dose of 8 mL was regularly administered at scheduled 45 min intervals for the PIEB mode regardless of body height. Furthermore, participants in the study group combined to use the PCEA mode of 6 mL of bolus dose in 10 min lockout intervals, and the maximum dose of PIEB + PCEA was limited to <30 mL/h. A rescue bolus dose was designed for intolerable pain initiated by the parturient in both groups. The rescue bolus regimen was 2 mg/mL of ropivacaine in a 10 mL volume, manually injected by the medical staff through the inserted epidural route.

During the period of painless labor, the vital signs of parturients were regularly monitored and recorded, including blood pressure, pulse oximetry, and heart rate. The physiological survey was carried out at 5 min intervals for the first 30 min, followed by 15 min intervals in the next 30 min, and then every 30 min interval until the end time point of labor analgesia. Hypotension was defined as a >20% reduction in systolic blood pressure from baseline.

### 2.3. Quality of Labor Analgesia and Outcomes

The clinical characteristics of the parturients, such as age, body weight and height, gestational age, gestational diabetes mellitus, and duration of epidural analgesia, were recorded. The quality of labor analgesia in the CEI and PIEB groups was compared. The primary outcomes were the time to T_1st_ and the number of manual rescue boluses. We adopted the Medical Research Council scale to evaluate the muscle power of the lower extremities bilaterally, and each score ranged from 0 to 5 accordingly.

0: No muscle activation;

1: Trace muscle activation, without achieving full range of motion;

2: Muscle activation with gravity eliminated, achieving full range of motion;

3: Muscle activation against gravity;

4: Muscle activation against some resistance;

5: Muscle activation against examiner’s full resistance.

Muscle power scores from the bilateral lower extremities were summed, and the lowest score of bilateral muscle power (LSMP) was monitored at any time point during labor. The summed muscle power scores at the time of delivery (ESMP) were also recorded. Furthermore, a VAS of 0–10 was self-appraised for the unidimensional measurement of pain intensity and progression. The status of muscle power and VAS score were measured and recorded every 30 min until the end of the study. Overall satisfaction with painless labor was assessed by the parturient herself at the end of labor analgesia. The satisfaction levels were as follows: 1, not satisfied; 2, slightly satisfied; 3, very satisfied; and 4, extremely satisfied. The outcome of the delivery process was evaluated as spontaneous delivery with or without being instrumentally assisted. Hypotension episodes and dysuria were also observed. The neonatal outcome was assessed using the Apgar score 1 and 5 min after birth and whether the newborn needed intensive care.

Techniques and parameters of the placement of the epidural catheter and doses of ropivacaine and fentanyl could be confounding variables. Therefore, the interspace level was punctured, and the cephalic depth of the threading catheter in the epidural space and total doses of ropivacaine and fentanyl were compared.

### 2.4. Statistical Analysis

The primary outcome was the cumulative percentage of primipara requiring a rescue dose during labor. A sample size calculation based on our pilot study showed that we required 305 participants to show the effect size = 0.178 of the cumulative percentage of primipara that required the rescue dose between the CEI and PIEB + PCEA groups and the assumption of power = 0.80, alpha = 0.05. Data in numeric variables were analyzed using the Kolmogorov–Smirnov test and expressed as medians (interquartile range (IQR]). Alternatively, the Mann–Whitney U test was used for non-parametric comparisons if numeric data were not normally distributed. Categorical variables were presented as numbers (%) and analyzed using the chi-square or Fisher’s exact test. The odds ratio (OR) of women who required clinician rescue more than twice in several variables was analyzed using a multiple binary logistic regression. A Kaplan–Meier survival analysis was used to evaluate the time to T_1st_ in both groups; *p* < 0.05 was indicated as statistically significant.

## 3. Results

A total of 570 parturients were enrolled in our study, of which 483 were finally included (Figure 1). Among the 483 women, 135 were nulliparous assigned to the CEI group and 348 to the PIEB + PCEA group. There were no significant differences between the two groups in the clinical characteristics, techniques, or parameters of epidural catheter placement; interspace level punctured; and cephalic depth of the threading catheter (Table 1).

T_1st_ in the PIEB + PCEA group was 30 min longer than that in the CEI group, with a significant difference (*p* < 0.005) (Table 2). The median [IQR] of T_1st_ was 210 [136–330] min in the PIEB + PCEA group and 180 [140–248] min in the CEI group. The total number of manual rescue boluses performed by the clinician was lower in the PIEB + PCEA group than in the control group. The median [IQR] rescue times were two [2,3,4] times in the CEI group and two [1,2,3] times in the PIEB + PCEA group. The difference between the two groups was statistically significant (*p* < 0.001). Although the average duration of labor analgesia was longer in the PIEB + PCEA group than that in the control group, the two groups did not show differences, with a *p*-value of 0.943. The cumulative percentage of primipara without any rescue dose during labor was compared between the two groups, and the data in the PIEB + PCEA group were relatively and sustainably higher (*p* < 0.05). At the end of the epidural analgesia administration, the cumulative percentage of unnecessary rescue was 7.8% in the PIEB + PCEA group and 1.5% in the control group. An analysis of anesthetic consumption during labor revealed no difference between the two groups.

The muscle power of the lower extremities was not reduced in either group. The vaginal delivery that required instrument assistance was 11.9% and 16.7% in the CEI and PIEB + PCEA groups, respectively (*p* = 0.187) (Table 2). Maternal blood pressure, heart rate, and oxygenation levels were comparable between the groups. The incidence of hypotension during epidural analgesia was 8.1% and 5.5% in the CEI and PIEB + PCEA groups, respectively (*p* = 0.272). After delivery, the questionnaires were retrieved from each patient. The rating levels of satisfaction, which were scored as four points (extremely satisfied), were compared between the two groups. The setting of PIEB + PCEA was superior to the control group, with a significant difference (*p* = 0.022). We also evaluated the impact of these two settings on neonatal outcomes at birth, including neonate body weight, Apgar score, and poor health that required intensive care. Our results showed that their performance was similar in these items (Table 3).

The time to T_1st_ was compared between both groups using the Kaplan–Meier survival analysis method. The time point called for first aid was extended in the PIEB + PCEA group compared to the CEI group, with positive significance (*p* < 0.001) (Figure 2).

In our study, the modality setting of the delivery devices was defined as poor performance if the participants in the group needed manual clinician rescue for more than two independent boluses. Dichotomous outcomes were compared using a multiple binary logistic regression method (Table 4). Our data revealed that T_1st_, total duration of labor analgesia, and CEI were risk factors for the poor performance of labor analgesia. Each 1 min increase in T_1st_ decreased the OR of poor performance by 0.99 times. Moreover, each 1 h increase in duration of analgesia increased the OR of poor performance by 1.01 times. The utilization of the CEI mode resulted in poor performance, with an OR of 2.635.

## 4. Discussion

Whether the modality setting for epidural delivery is superior remains debatable [18]. This study showed that the PIEB + PCEA setting for epidural analgesia, compared to the CEI setting, significantly reduced manual rescue by the clinician and extended the time required for T_1st_. The PIEB + PCEA setting performed better than the CEI setting and took advantage of the parturients and clinician staff. Using the PIEB + PCEA setting can reduce the frequency of breakthrough pain (OR, 0.35; 95% confidence interval [CI], 0.19–0.62). A previous study reported that the percentage of breakthrough pain or manual rescue intervention by anesthesia staff ranged from 10% to 64% [5,6,7,9,13,15,19]. In a study by Wong et al. the percentage of breakthrough pain was up to 50% [15]. The manipulation of anesthetic regimens, concurrent use of CSE, background infusion dose, interval, and volume of programmed intermittent bolus influenced the quality of labor pain relief. Several studies concluded that PIEB was superior to CEI in reducing the frequency of interventions [5,6,9,13]. Although they had different manipulations in their research, the frequency of interventions was not as low as zero. Currently, there is no solid evidence that CEI settings result in a lower incidence of rescue interventions than PIEB settings [7,14,19]. Many previous studies have reported contradictory comparisons of the incidence of rescue and T_1st_ between the PIEB and CEI settings. Some studies have revealed that the T_1st_ in the PIEB setting was longer than that in the CEI setting [9,14]. However, McKenzie et al. showed a similar T_1st_ between them, with no statistical difference, although the T_1st_ in the CEI group was double that in the PIEB group [13]. In their study by McKenzie et al. labor analgesia was initiated with epidural bupivacaine or CSE with intrathecal bupivacaine. This may have contributed to the disagreement with others [13]. Our study excluded patients who concurrently received CSE for labor pain. Regarding novelty, we tested the efficiency and superiority of using a combination of PIEB and PCEA settings referencing the CEI setting. Our results showed that the PIEB + PCEA setting took advantage of a better T_1st_ and contributed to fewer manual rescue times than the CEI group, which had a shorter labor period and was supposed to experience fewer rescues in our study.

Some studies have reported that maternal satisfaction was better in PIEB settings than in CEI settings [5,6,7]. In other trials, there were no significant differences in pain scores between PIEB and CEI [5,9,14,20]. Our present study revealed that the use of the PIEB + PCEA setting for labor analgesia provided better maternal satisfaction and a reduced workload for medical staff compared to the CEI setting. In our design, a maternal satisfaction survey was adopted for quality measurement instead of the numerical rating scale (NRS) at the end of labor. The satisfaction level did not coincide with pain control, since satisfaction depends on reliance, reassurance, duration of labor, and so on. The survey was conducted independently by the patients and reflected on the overall quality of labor analgesia. It would contain a subjective evaluation of NRS, muscle power weakness, and undesirable effects.

Breakthrough pain is difficult to manage with labor analgesia. Regardless of the modality, an instant manual rescue bolus is a convincing way to relieve intolerant pain if the epidural catheter is already in situ. The settings of background infusion and intermittent boluses are designed to ameliorate the necessity of manual rescue boluses, not only to save manpower but also to improve the quality of labor analgesia. As breakthrough pain occurs, preparation for manual rescue and the onset of epidural anesthetics take time. This usually leads to unendurable labor pain and poor satisfaction with parturients. Consequently, less manual rescue time results in a high quality of labor analgesia and may contribute to high maternal satisfaction [5,6]. Our results showed that the PIEB + PCEA group benefited from good maternal satisfaction and experienced fewer manual rescue boluses. When comparing performance using the regression model, the use of the CEI setting was inferior to the PIEB + PCEA setting, with a higher percentage of rescue interventions more than twice, with an OR of 2.619. Some authors have reported that PIEB prolongs the duration of analgesia relative to CEI [9,14], provides better or similar pain relief [9,13,14], or has less motor block [12,21,22]. Other studies have not shown any difference in motor block [9] or an unexpected cesarean section between PIEB and CEI [5,9,14]. In our report, the motor block at the end of labor analgesia and the lowest degree of motor function during the whole course were not different between the two groups. Importantly, parturients in the PIEB + PCEA group were more satisfied than those in the CEI group, which is consistent with previous studies [6].

Our study design differed from that of other studies; we combined the dual setting of PIEB + PCEA in one pump equipped with one bag of anesthetic solution. In the past, this dual-setting design required collaboration between two pumps connected to an epidural catheter using a three-way hub connector [6,15] or using computer-assisted pumps [7,10]. The labor and delivery unit are often part of a busy and urgent scenario; however, our design of dual setting in one pump provided a simple and easy-to-use manner and achieved safe and good satisfaction. Furthermore, the cost of one bag of the analgesic solution is lower than that of two pumps with two bags, not to mention the small number of anesthesiologists in the COVID-19 pandemic.

Despite manipulations in a time interval or volume of intermittent bolus and anesthetic selection, previous research did not reach a consensus to show whether the PIEB setting took advantage of T_1st_ and manual rescue more than the CEI setting [9,14,15,22]. Interestingly, we found that the performance of both the T_1st_ and manual rescue boluses was positive and significant in the PIEB + PCEA group. The delivery pump that we used was a commercial device (CADD Solis Epidural Pump, Smiths Medical, St. Paul, MN, USA) and was capable of dual settings (PIEB plus PCEA). The default rate of the delivery impulse can reach 175 mL/h. The delivery rate adopted in previous articles was different and was reported to be 75–400 mL/h. Therefore, the intermittent bolus modality produced high pressure and spread the anesthetic solution in the epidural space despite the volume of the bolus [15]. Compressed high-speed injection could alter the pharmacokinetics and level of analgesia referred to by constant low-pressure infusion [20]. However, many studies have disagreed with this hypothesis that the delivery impulse could affect the painless duration, manual rescue doses, and even sensory level [7,9,10,14,19]. Delgado et al. showed that the high flow rate at 500 mL/h contributed to a lower bolus requirement, similar to the rate at 250 mL/h in another study [23]. If a high-compressed impulse leads to a good performance of the PIEB, the conclusion remains elusive.

An anesthetic solution containing 0.625–1 mg/mL of ropivacaine and 2 μg/mL of fentanyl was the most common regimen for maintenance. A high concentration of local anesthetics does not take it for granted, resulting in good-quality analgesia. A previous study revealed that the CEI group infused with 2 mg/mL of ropivacaine resulted in more undesirable motor blocks than the PIEB group administered a low concentration of ropivacaine (1 mg/mL) [12]. Our study used ropivacaine (0.625 mg/mL) plus fentanyl (2 μg/mL) for background infusion in the PIEB setting, which resulted in a longer T_1st_, fewer manual boluses, and better analgesia than those in the CEI setting. The PIEB + PCEA modality had a limited incidence of motor block or adverse neonatal outcomes.

The limitations of the present study are that we lacked the comparison with an extra group of PCEA+CEI. Readers may argue that if PCEA was added to the CEI group, the conclusion may be redefined. However, this hypothesis needs further evaluation in the future. This cohort study did not involve random allocation and was based on the clinical judgment of each obstetric anesthesia attending physician. Therefore, the research evidence may not be directly comparable to that of randomized controlled trials. The limited number of cases in the CEI group may introduce potential bias into the results.

## 5. Conclusions

In summary, we found that the PIEB + PCEA group may have a longer duration for the first requested manual rescue, fewer manual rescue boluses, better satisfaction, and no significant increase in adverse events compared to the CEI group. The future directions regarding research in this territory may be on the associations with labor length, different modes of delivery, or cardiotocography tracing.

## Figures and Tables

**Figure 1 healthcare-11-01350-f001:**
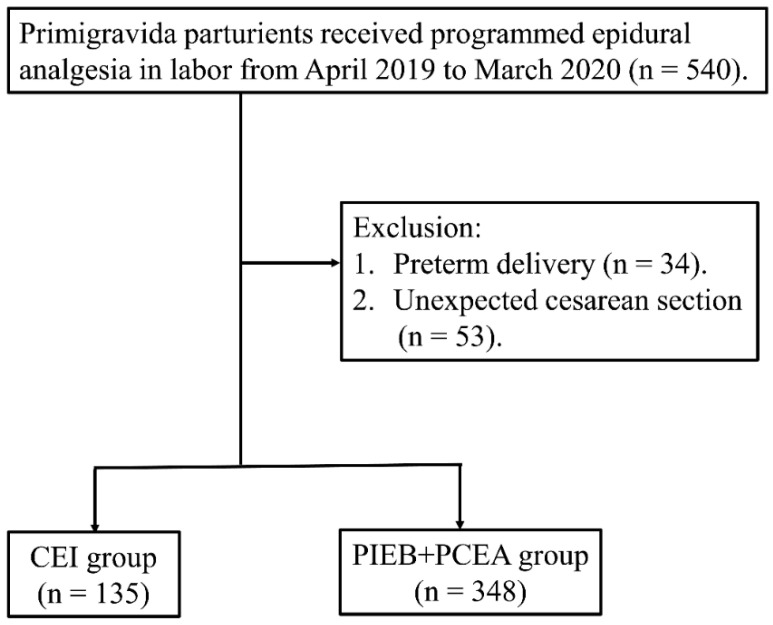
Flow diagram of the study participants. CEI, continuous epidural infusion; PIEB, programmed intermittent epidural boluses; PCEA, patient-controlled epidural analgesia.

**Figure 2 healthcare-11-01350-f002:**
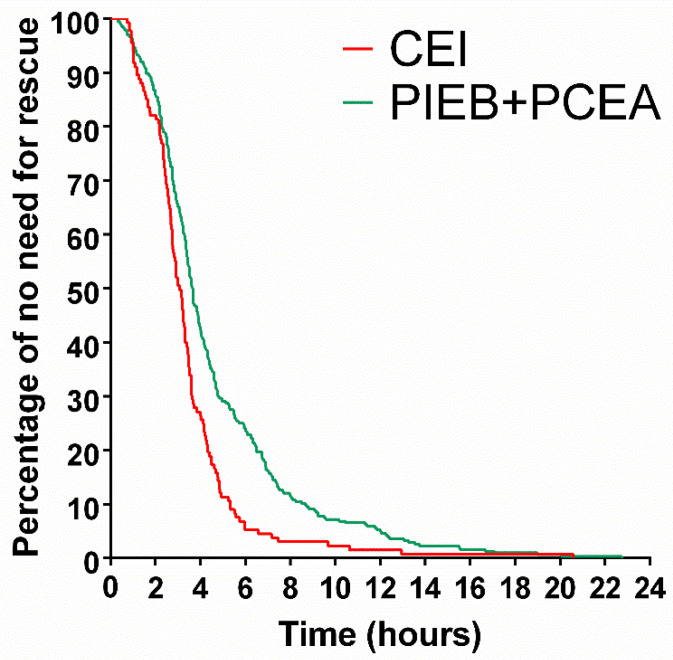
Cumulative percentage of primipara without rescue dose between the CEI and PIEB + PCEA groups. The overall primipara percentage without rescue was compared between the CEI and PIEB + PCEA groups. CEI, continuous epidural infusion; PIEB, programmed intermittent epidural boluses; PCEA, patient-controlled epidural analgesia.

**Table 1 healthcare-11-01350-t001:** Demographic data and clinical characteristics of parturients who received PIEB + PCEA and CEI for labor analgesia.

	CEI (n = 135)	PIEB (n = 348)	*p*-Value
Age (years)	31 (28–34)	31 (28–33)	0.824
Weight (kg)	66 (61–73)	66 (61–74)	0.928
Height (cm)	160 (157–163)	160 (156–163)	0.166
Gestational age (weeks)	39.4 (38.6–40.2)	39.3 (38.6–40.0)	0.153
Gestational diabetes mellitus	2 (1.5%)	3 (0.9%)	0.546
Epidural insertion site			
L2–3	0.7%	2.6%	0.296
L3–5	99.3%	97.4	
Cephalic depth of threading catheter (cm)			
≤5	16.3%	23.6%	0.066
5.1–7	83.0%	76.4%	
≥7.1	0.7%	0%	
Fixed depth on the skin (cm)			
≤10	40.7%	37.1%	0.363
10.1–13	59.3%	61.8%	
≥13.1	0.0%	1.1%	

Data are presented as median (IQR) or number (%). CEI, continuous epidural infusion; PIEB, programmed intermittent epidural bolus; PCEA, patient-controlled epidural analgesia.

**Table 2 healthcare-11-01350-t002:** Quality of labor analgesia in the CEI and PIEB groups.

	CEI (n = 135)	PIEB (n = 348)	*p*-Value
Number of rescues	2 (2–4)	2 (1–3)	<0.001 *
Maternity without rescue dose	1.5%	7.8%	0.009 *
Time to first manual rescue (mins)	180 (140–248)	210 (136–330)	0.010 *
Total duration of labor analgesia (mins)	368 (256–565)	403 (229–650)	0.943
Instrumental delivery	11.9%	16.7%	0.187
LSMP	10 (10–10)	10 (10–10)	0.187
ESMP	10 (10–10)	10 (10–10)	0.553
Hypotension episode	11 (8.1%)	19 (5.5%)	0.272
Extremely satisfied	23.7%	34.7%	0.022 *
Ropivacaine dose (mg)	49.5 (33.8–70.6)	49.2 (31.6–79.9)	0.461
Fentanyl dose (μg)	128.8 (89.6–197.6)	147.6 (88.0–238.6)	0.601

Data are presented as number (%) or median (IQR), and * *p* < 0.05, indicating statistical significance. CEI, continuous epidural infusion; PIEB, programmed intermittent epidural boluses; PCEA, patient-controlled epidural analgesia; LSMP, the lowest summed score of bilateral lower extremities muscle power during labor; ESMP, summed score of bilateral lower extremities muscle power at the time of delivery.

**Table 3 healthcare-11-01350-t003:** Outcomes of neonatal at birth.

	CEI (n = 135)	PIEB (n = 348)	*p*-Value
Body weight (g)	3119.4 ± 339.9	3089.8 ± 328.6	0.379
Apgar score (at birth)	9 (9–9)	9 (9–9)	0.347
Apgar score (5 min after birth)	10 (10–10)	10 (10–10)	0.124
Require intensive care	2 (1.5%)	5 (1.4%)	0.971

The Apgar score comprises five components: color, heart rate, reflexes, muscle tone, and respiration, each of which is given a score of 0, 1, or 2 [17]. Data are presented as numbers (%) or medians (IQR). CEI, continuous epidural infusion; PIEB, programmed intermittent epidural bolus; PCEA, patient-controlled epidural analgesia.

**Table 4 healthcare-11-01350-t004:** Binary logistic regression analysis for the need of clinician rescue more than twice.

Variable	OR	95% CI of OR	*p*-Value
Age (years)	0.969	0.908–1.034	0.377
Weight (kg)	1.013	0.986–1.041	0.348
Height (cm)	1.003	0.945–1.064	0.930
Gestational age (week)	1.015	0.747–1.379	0.925
Epidural insertion at L3–5	2.870	0.182–4.565	0.909
Time to the first manual rescue dose (min)	0.990	0.988–0.992	0.001 *
Total duration of labor analgesia	1.010	1.008–1.012	0.001 *
Dysuria	1.507	0.174–13.082	0.710
CEI setting	2.635	1.491–4.655	0.001 *

* *p* < 0.05 represents statistical significance. CEI, continuous epidural infusion; OR, odds ratio.

## Data Availability

No new data were created.

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
