# Peer review of "Combined Programmed Intermittent Bolus and Patient-Controlled Bolus Is a More Favorable Setting for Epidural Pain Relief Than Continuous Infusion"

_healthcare, 2023, doi:10.3390/healthcare11091350_

Round 1

Reviewer 1 Report

Your study about labor anaestesia deals with an important issue due to its clear benefit for all women in labor. Congratulations for your interest and effort in this field.

Muscle power score that is mentioned in Methods section requires a clear description.

You mentioned that "The quality of the delivery process was evaluated as cesarean section or vaginal delivery and instrumentally assisted or spontaneous delivery" - you need to rephrase or clearly specify what does quality of delivery process means. CS vs vaginal delivery is not correlated with epidural analgesia. Instrumental delivery is in some studies associated with epidural.

You mentioned that all associated pathology were excluded ("Parturients were healthy without systemic disease"), but you mention in table 1 - GDM which is a medical condition which can sometimes change obstetrical practice.

From table 3 it is not clear how Apgar scores were assesed. All your group had Apgar 10 at 5 minutes but also some neonatal intensive care was needed? 

Conclusion section can be extended with a take home message and some future directions regarding research - eg labor length association, mode of delivery, CTG tracing associations.

Author Response

Reviewer #1:

Your study about labor anaesthesia deals with an important issue due to its clear benefit for all women in labor. Congratulations for your interest and effort in this field.

ANS: We sincerely thank for your comments.

  • Muscle power score that is mentioned in Methods section requires a clear description.

ANS: We added more information to descript the score system in Page 3, Line 121-129 as follow.

“We adopted the Medical Research Council scale to evaluate the muscle power of the lower extremities bilaterally, and each score ranged from 0 to 5 accordingly.

0: No muscle activation

1: Trace muscle activation, without achieving full range of motion

2: Muscle activation with gravity eliminated, achieving full range of motion

3: Muscle activation against gravity

4: Muscle activation against some resistance

5: Muscle activation against examiner’s full resistance”

  • You mentioned that "The quality of the delivery process was evaluated as cesarean section or vaginal delivery and instrumentally assisted or spontaneous delivery" - you need to rephrase or clearly specify what does quality of delivery process means. CS vs vaginal delivery is not correlated with epidural analgesia. Instrumental delivery is in some studies associated with epidural.

ANS: Thanks for your kind reminder. Unexpected cesarean section is an exclusion criterion in our study. We re-wrote the sentences in Page 3, Line 138-139 as follow. “The outcome of the delivery process was evaluated as spontaneous delivery with or without instrumentally assisted.”

  • You mentioned that all associated pathology were excluded ("Parturients were healthy without systemic disease"), but you mention in table 1 - GDM which is a medical condition which can sometimes change obstetrical practice.

ANS: We re-edited the sentences in Page 2, Line 86 as follow. “Parturients were healthy without systemic disease before getting pregnant,”

  • From table 3 it is not clear how Apgar scores were assesed. All your group had Apgar 10 at 5 minutes but also some neonatal intensive care was needed?

ANS: (1) We added the explanation and cite a reference (The Apgar Score. Pediatrics. 2015,136, 819–822.) for this scoring system on the footnote of Table 3 in Page 6 as follow. “The Apgar score comprises five components: color, heart rate, reflexes, muscle tone, and respiration, each of which is given a score of 0, 1, or 2 (Pediatrics 2015, 136, 819-822, doi:10.1542/peds.2015-2651).” (2) Apgar scores were recorded at birth and 5 min after birth. These neonates needed intensive cares were not due to poor Apgar score. The Babies with breathing trouble or heart problems in later, birth defects, or infections were also cared for in the NICU. Other high-risk factors would be an indication including maternal diabetes, bleeding, sexually transmitted diseases, too little or too much amniotic fluid, premature rupture of membranes, breech birth or other abnormal position, baby's first stool (meconium) passed during pregnancy into the amniotic fluid, umbilical cord wrapped around the baby's neck, forceps delivery.

  • Conclusion section can be extended with a take home message and some future directions regarding research - eg labor length association, mode of delivery, CTG tracing associations.

ANS: Thank you. We re-edited the conclusion in Page 9, Line 334-337 as follow. “The future directions regarding research on this territory may be on the associations with labor length, different mode of delivery, or cardiotocography tracing.”

Reviewer 2 Report

The authors have hypothesised that PIEB + PCEA is better than CEI for labour analgesia in terms of manual rescue doses required and satisfaction.

One of the major drawbacks of the study is tha absence of PCEA with the CEI. Why can’t the two be combined? CEI had a higher infusion rate as compared to programmable boluses. True there are studies which have reported better outcomes with the latter with certain disadvantages of the former namely motor block and increased instrumentation and more drug, the same have not been brought out by this study.

also, the results presented are statistically, but not clinically relevant. The modest increase of one extra manual bolus… would not impact the load. Also if PCEA is added to CEI, it might even prove better than PIEB.

Limitations of the study are lacking, and going by the Kaplan Meir graph, the probability of manual bolus seems less with CEI.
Rather than absolute statements, in view of the lack of reasonable clinical relevance, the conclusion may be toned down a bit.

Author Response

Reviewer #2:

The authors have hypothesised that PIEB + PCEA is better than CEI for labour analgesia in terms of manual rescue doses required and satisfaction.

ANS: We sincerely thank for the valuable comments.

  • One of the major drawbacks of the study is the absence of PCEA with the CEI. Why can’t the two be combined? CEI had a higher infusion rate as compared to programmable boluses. True there are studies which have reported better outcomes with the latter with certain disadvantages of the former namely motor block and increased instrumentation and more drug, the same have not been brought out by this study.

ANS: There are many studies have reported the differences between PIEB and CEI groups. We thought added a same variate (PCEA) to each group (PIEB+PCEA v.s. CEI+PCEA) may not alter previous conclusion, but damp the differences between groups. Other reason is that group of PIEB+PCEA or CEI only needed one machine to complete, but CEI+PCEA needed two machine in each one patient. However, we do not know if PCEA is added to CEI provides a comparable effect to PIEB+PCEA. It needs further evaluation in future.

  • also, the results presented are statistically, but not clinically relevant. The modest increase of one extra manual bolus… would not impact the load. Also if PCEA is added to CEI, it might even prove better than PIEB.

ANS: (1) Our data revealed a statistical difference in manual bolus needed. In our hospital, there are more than 5 parturients every day. Beside of these parturients, the medical stuff needs to take care others who were not parturient. The reduce of one extra manual bolus for every parturient would improve total work load. (2) Yes, it is possible that if PCEA was added to CEI, the conclusion may be redefined.

  • Limitations of the study are lacking, and going by the Kaplan Meir graph, the probability of manual bolus seems less with CEI. Rather than absolute statements, in view of the lack of reasonable clinical relevance, the conclusion may be toned down a bit.

ANS: (1) We added the limitation of our study in Page 9, Line 330-332 as follow. “The limitations of present study are lacking the comparison with an extra group of PCEA+CEI. Readers may argue with that if PCEA was added to CEI group, the conclusion may be redefined. However, this hypothesis needs further evaluation in future.” (2) Thank you for the nice suggestions. The conclusion is toned down a bit in Page 9, Line 333-334 as follow. “In summary, we found that the PIEB+PCEA group may have a longer duration for the first requested manual rescue, fewer manual rescue boluses, better satisfaction, and no significant increase in adverse events compared to the CEI group.”

Reviewer 3 Report

This is my more specific comment.  

"This is an essential and well-written paper on Epidural Pain Relief during vginal delivery. Overall the report found that Programmed Intermittent epidural bolus with the PCEA setting as a maintenance infusion had a longer duration for the first manual rescue boluses, excellent satisfaction, and no significant increase in adverse events compared to the CEI setting. 

I enjoyed reading this paper, which provides critical information on epidural pain relief for vaginal delivery.

 I only have a couple of minor comments: 

- was this study conducted in one hospital or many in many facilities? How many anesthesiologists take part in this study? Was one nurse sign to one patient - a woman giving birth?  What determined the extra bolus doses of epidural analgesia? Was it a VAS score?  

- Which resulted in the patients being included in the CEI group? Why are these two groups not equal?  

- references- why are all of them older than five years? There are some excellent new trials considering epidural anesthesia in vaginal delivery. I highly recommend that you include some of them in your manuscript."

Author Response

Reviewer #3: "This is an essential and well-written paper on Epidural Pain Relief during vaginal delivery. Overall the report found that Programmed Intermittent epidural bolus with the PCEA setting as a maintenance infusion had a longer duration for the first manual rescue boluses, excellent satisfaction, and no significant increase in adverse events compared to the CEI setting.

I enjoyed reading this paper, which provides critical information on epidural pain relief for vaginal delivery.

I only have a couple of minor comments:

  • - was this study conducted in one hospital or many in many facilities? How many

anesthesiologists take part in this study? Was one nurse sign to one patient - a woman giving birth? What determined the extra bolus doses of epidural analgesia? Was it a VAS score?

ANS: This study was conducted at a single tertiary hospital in a medical center. Three anesthesiologists specializing in obstetric anesthesia participated in the study. One anesthesia nurse was responsible for providing nursing care to multiple parturients and for some procedures in the operating room. The administration of additional bolus doses was determined by various clinical factors, including pain scores, analgesic dermatomes, patient tolerance, fetal heart rate, and maternal blood pressure.

  • - Which resulted in the patients being included in the CEI group? Why are these two groups not equal?

ANS: Patient allocation was determined based on the attending anesthesiologist responsible for obstetric anesthesia on the day of delivery, as well as the progress of labor, individual experience, and medical preferences. There were no statistically significant differences in basic clinical and physiological data, such as age, gender, height, weight, injection site depth, between the two groups of patients, as shown in Table 1.

  • - references- why are all of them older than five years? There are some excellent new trials considering epidural anesthesia in vaginal delivery. I highly recommend that you include some of them in your manuscript."

ANS: Thank you for the good suggestion. We cited some recent studies in our manuscript (Korean J Anesthesiol. 2019 Oct;72(5):472-478. doi: 10.4097/kja.19156. and Can J Anaesth. 2023 Mar;70(3):406-442.).

Reviewer 4 Report

- Was 'Fidkowski CW, Shah S, Alsaden MR. Programmed intermittent epidural bolus as compared to continuous epidural infusion for the maintenance of labor analgesia: a prospective randomized single-blinded controlled trial. Korean J Anesthesiol. 2019 Oct;72(5):472-478. doi: 10.4097/kja.19156. Epub 2019 Jun 20. PMID: 31216846; PMCID: PMC6781207.' reviewed for references and discussion? 

-  Clear methodology but not novel. This article adds to the body of evidence supporting PIEB with PCEA

Author Response

Reviewer #4:

  • - Was 'Fidkowski CW, Shah S, Alsaden MR. Programmed intermittent epidural bolus as compared to continuous epidural infusion for the maintenance of labor analgesia: a prospective randomized single-blinded controlled trial. Korean J Anesthesiol. 2019 Oct;72(5):472-478. doi: 10.4097/kja.19156. Epub 2019 Jun 20. PMID: 31216846; PMCID: PMC6781207.' reviewed for references and discussion?

ANS: Thank you for the recommendations. We cited this reference in discussion section in Page 2, Line 45-47 as follow. “Fidkowski et al. reported that PIEB regimens at 10 ml every 60 min for labor analgesia decreased breakthrough pain and manual rescue compared by CEI.”

  • - Clear methodology but not novel. This article adds to the body of evidence supporting PIEB with PCEA

ANS: Thank you for the kind commends.

Reviewer 5 Report

The paper is interesting and well written, however I have some concerns about the methods that are not so clear to me. Was it an observational study, a retrospective one or what? What criterion was used to divide the population in continuous vs intermittent bolus? were the rescue doses  similar (concentration and volume) in both groups?

were the 2 groups stratified for different variables (age, duration of labor for example) to control if they were comparable?

It is not fully clear moreover how the difference in rescue doses could be significant since the median is 2 in both groups...maybe  it could be more useful to add the number of patient asking for a rescue dose.

Maybe it would be useful to add this very recent meta-analysis (Wydall S, Zolger D, Owolabi A, Nzekwu B, Onwochei D, Desai N. Comparison of different delivery modalities of epidural analgesia and intravenous analgesia in labour: a systematic review and network meta-analysis. Can J Anaesth. 2023 Mar;70(3):406-442. )

The satisfaction level does not coincide with pain control, since satisfaction depends on reliance, reassurance, duration of labor, and so on.

Moreover, the effect of epidural analgesia depend on volume as well as concentration of the mixture used, so the conclusion about the effect of spread, volume and pressure in intermittent modality should be better discussed.

Author Response

Reviewer #5: The paper is interesting and well written, however I have some concerns about the methods that are not so clear to me.

  • Was it an observational study, a retrospective one or what? What criterion was used to divide the population in continuous vs intermittent bolus? were the rescue doses similar (concentration and volume) in both groups?

ANS: This is a prospective cohort study with enrolled cases, and the results were analyzed retrospectively. Patient allocation was determined based on the attending anesthesiologist responsible for obstetric anesthesia on the day of delivery, as well as the progress of labor, individual experience, and medical preferences. The rescue dose was standardized and described in section 2.2 "Modality Settings of the Delivery Device for Labor Analgesia".

  • were the 2 groups stratified for different variables (age, duration of labor for example) to control if they were comparable?

ANS: Patient allocation was determined based on the attending anesthesiologist responsible for obstetric anesthesia on the day of delivery, as well as the progress of labor, individual experience, and medical preferences. There were no statistically significant differences in basic clinical and physiological data, such as age, gender, height, weight, injection site depth, between the two groups of patients, as shown in Table 1.

  • It is not fully clear moreover how the difference in rescue doses could be significant since the median is 2 in both groups...maybe it could be more useful to add the number of patients asking for a rescue dose.

ANS: The number of rescues was non-normally distributed, and based on non-parametric analysis, there was a significant difference in the number of rescues between the two groups (the median values were both 2, but the interquartile ranges were 2-4 v.s. 1-3). This suggests that at least 50% of the patients in the PIEB+PCEA group required one less rescue compared to the CEI group. Meanwhile, the PIEB+PCEA group had a higher percentage of patients who did not require any rescue doses compared to the CEI group.

  • Maybe it would be useful to add this very recent meta-analysis (Wydall S, Zolger D, Owolabi A, Nzekwu B, Onwochei D, Desai N. Comparison of different delivery modalities of epidural analgesia and intravenous analgesia in labour: a systematic review and network meta-analysis. Can J Anaesth. 2023 Mar;70(3):406-442. )

ANS: Thank you for the nice recommendations. We cited this reference in discussion section in Page 7, Line 230.

  • The satisfaction level does not coincide with pain control, since satisfaction depends on reliance, reassurance, duration of labor, and so on.

ANS: Thank you for the suggestions. We re-edited our manuscript in Page 7, Line 264-265 as follow. “In our design, a maternal satisfaction survey was adopted for quality measurement instead of the numerical rating scale (NRS) at the end of labor. The satisfaction level does not coincide with pain control, since satisfaction depends on reliance, reassurance, duration of labor, and so on. The survey was conducted independently by the patients and reflected on the overall quality of labor analgesia. It would contain a subjective evaluation of NRS, muscle power weakness, and undesirable effects.”

  • Moreover, the effect of epidural analgesia depends on volume as well as concentration of the mixture used, so the conclusion about the effect of spread, volume and pressure in intermittent modality should be better discussed.

ANS: (1) Yes, volume and concentration of the mixture are the important variables, and have effects on epidural analgesia. However, our study was not focused on volume and concentration, and we used the same concentration of analgesic for maintenance dosages in 2 groups. Besides, the mean total dosages of ropivacaine in our 2 groups were also equal (49.5 mg and 49.2 mg, p=0.461). (2) We deleted the sentences in Page 8, Line 315-320 as follow. “Some may argue that the higher anesthetic volume in PIEB than that in CEI contributes to better performance in labor analgesia. However, many published articles used equal or less volumes of anesthetic in the PIEB group and achieved better performance than the CEI group [6,8,9]. This evidence concludes that the strengths of PIEB settings in labor pain relief are not solely attributed to pressure, volume, or spread of the anesthetic.

Round 2

Reviewer 5 Report

The paper is much improved. Please just specify in methods how patients were allocated in 2 groups and add in limits the low sample size, the non- randomised nature of the study, the lack of blinding, that all may represent factors  of  increased bias.

Author Response

Comments and Suggestions for Authors

The paper is much improved. Please just specify in methods how patients were allocated in 2 groups and add in limits the low sample size, the non- randomized nature of the study, the lack of blinding, that all may represent factors of increased bias.

ANS: We sincerely thank for your valuable comments. We added the details of allocation in the section of Method Line 98-101 and the potential bias of this non-randomized, non-blinding in the section of limitation Line 329-333.

Line 98-101 “Participants allocation was determined based on the attending anesthesiologist responsible for obstetric anesthesia on the day of delivery, as well as the progress of labor, individual experience, and medical preferences.”

Line 329-333 “This cohort study did not involve random allocation and was based on the clinical judgment of each obstetric anesthesia attending physician. Therefore, the research evidence may not be directly comparable to that of randomized controlled trials. The limited number of cases in the CEI group may introduce potential bias into the results. “